# Forgetting: A New Mechanism Towards Better Large Language Model Fine-tuning

**Ali Taheri**[*]
*Max Planck Institute for Software Systems*

**Alireza Taban**[*]
*Isfahan University of Technology*

**Qizhou Wang**
*Hong Kong Baptist University*

**Shanshan Ye**[†]
*University of Technology Sydney*

**Abdolreza Mirzaei**
*Simon Fraser University*

**Tongliang Liu**
*The University of Sydney*

**Bo Han**
*Hong Kong Baptist University*

**Reviewed on OpenReview:** *https://openreview.net/forum?id=s36smEoUoX*

## Abstract

Supervised fine-tuning (SFT) plays a critical role for pretrained large language models (LLMs), notably enhancing their capacity to acquire domain-specific knowledge while preserving or potentially augmenting their general-purpose capabilities. However, the efficacy of SFT hinges on data quality as well as data volume, otherwise it may result in limited performance gains or even degradation relative to the associated baselines. To mitigate such reliance, we suggest categorizing tokens within each corpus into two parts—**positive** and **negative** tokens—based on whether they are useful to improve model performance. Positive tokens can be trained in common ways, whereas negative tokens, which may lack essential semantics or be misleading, should be explicitly forgotten. Overall, the token categorization facilitates the model to learn less informative messages, and the forgetting guides the model on what information to learn more precisely. We conduct experiments across diverse and well-established benchmarks using various model architectures, demonstrating that this forgetting mechanism enhances model performance.

## 1 Introduction

In recent years, we have witnessed emerging advancements in large language models (LLMs) (Brown et al., 2020; Achiam et al., 2023), powered by transformer-based architectures (Vaswani et al., 2017) with billions of parameters and extensive pre-training on trillions of tokens (Zhao et al., 2023). These models have evolved rapidly with continuous improvements in architectural design, training strategies, and scaling

---

[*]Equal contribution.
[†]Corresponding author.

techniques (Hoffmann et al., 2022). They exhibit exceptional performance across a wide range of complex linguistic tasks, including reasoning, solving mathematics (Shao et al., 2024), summarization (Nallapati et al., 2016), language understanding, code generation (Chen et al., 2021; Jiang et al., 2026), question answering (Rajpurkar et al., 2016), etc.

Although powerful, LLMs still require SFT to enhance their performance in specialized tasks (Chung et al., 2024; Aggarwal et al., 2024; Strangmann et al., 2024; Lialin et al., 2023). SFT typically involves adapting the current LLM using conditional maximum likelihood principles on fine-tuning data comprising prompt-response pairs. However, its success heavily relies on the quality and volume of the data: Low quality can mislead the model learning (Dodge et al., 2021; Luccioni & Viviano, 2021; Welbl et al., 2021; Longpre et al., 2024), introducing biases or inaccuracies that degrade performance, and small-scale datasets will hinder the model's ability to generalize well (Ghosh et al., 2024). On the other hand, collecting the ideal data needed for SFT can be challenging in practice. Generally speaking, task-specific data are often scarce, particularly in niche or emerging domains (Ghosh et al., 2024; Ma et al., 2024), making it difficult to collect a sufficiently diverse dataset. Additionally, ensuring data quality is a non-trivial task, as it involves curating examples that are both representative and free from noise or errors. Even for humans, identifying whether the data meet high-quality standards can be difficult due to the subtleties of language and context. Consequently, the lack of high-quality, task-specific data becomes a bottleneck for SFT, limiting the potential of LLMs to excel in specialized applications.

**How can we mitigate the impacts of data on fine-tuning?** Data filtering (Albalak et al., 2024) offers a promising solution. Specifically, it involves selecting a subset of data from the whole set that is expected to be more beneficial for the targeted LLM than the original. With proper selection rules, such as gradient behaviors (Albalak et al., 2023), margins, loss, and influence (Bejan et al., 2023), filtering can refine data quality effectively. However, this comes at the cost of reducing the scale of the dataset, raising open questions about the trade-off between quality and scales and its impact on the generalization of the resulting model. Existing literature has attempted to mitigate this issue by exploring data rephrasing (Eldan & Russinovich, 2023; jin et al., 2024), while this approach heavily depends on manual efforts and/or expensive generators that are task-specific.

In this paper, we explore a new mechanism towards better LLM fine-tuning, referred to as **forgetting**. Following previous wisdom (Yuan et al., 2024; Eldan & Russinovich, 2023; Wang et al., 2025b; Koh & Liang, 2017), we begin by performing data filtering at the token level, categorizing tokens as either **positive** or **negative** based on their influence to enhancing performance. Note that token-level filtering helps preserve the data scale as much as possible, thus adopting as a default choice. Then, for positive tokens, conditional maximum likelihood is applied as usual, since our selection rules ensure that their learning will benefit the current model. Furthermore, for negative tokens, rather than simply discarding them, we propose applying forgetting (also referred to as unlearning (Li et al., 2025; De Cao et al., 2021; Jang et al., 2023; Maini et al., 2024; Yao et al., 2024b; Wang et al., 2025b)) to reduce the likelihood of their generation. Compared to positive ones, negative tokens are more likely to carry uninformative or even misleading knowledge. Explicitly forgetting these tokens not only prevents the model from generating them but also helps avoid overfitting to the current corpus. Moreover, we maintain the same data scale as in conventional fine-tuning, while taking some data (tokens more accurately) as negative samples during training to facilitate model generalization.

Although straightforward to implement, we demonstrate the importance of forgetting in SFT for improved generalization through our extensive experiments. Specifically, we build our training corpus across 5 representative reasoning, knowledge and conversational datasets, and evaluate our forgetting mechanism alongside baseline methods on 5 diverse benchmark datasets, incorporating various LLMs as base models. For example, as shown in Table 3 in Section 5, using LLaMA3.2-1B as the base model, our approach achieved a 2.51% improvement over that without forgetting and a 4.49% improvement over the fine-tuned model on full tokens. Similarly, with LLaMA3.2-3B, we obtained a 3.4% improvement over that without forgetting and 5.28% over fine-tuned model on full tokens. Additionally, with LLaMA3.1-8B, our approach resulted in a 4.21% improvement over the no forgetting approach, and a 8.25% improvement over the fine-tuned model on full tokens. To validate scalability to larger model sizes, we conducted experiments with LLaMA-2-13B in Appendix B.1, confirming the forgetting mechanism's generalization capability across different

scales. Furthermore, we demonstrate our effectiveness across other model architectures (Qwen2.5-3B and GPT-Neo-2.7B) and diverse evaluation tasks in Appendix B.3.

**Connection with broader literature.** The mechanism of forgetting is closely connected to preference optimization (PO) (Rafailov et al., 2023). Recalling that, many representative PO methods, such as direct preference optimization (DPO) (Rafailov et al., 2023) and proximal policy optimization (PPO) (Schulman et al., 2017), can broadly be reviewed as combining the objectives of learning and forgetting. They aim to increase the likelihood of generating preferred corpora while reducing that of the dispreferred one. However, these methods are derived from the original PO objectives, which are inherently tied to problem setups and rely on manual labeling or reward models for preference annotation. In contrast, we focus on the SFT problems, where the forgetting mechanism acts as an enhancement strategy rather than a indispensable component of the problem formulation. Our method is inspired by PO but more focuses on the mechanism of forgetting as an integral component within learning. This approach helps mitigate the negative effects of low-quality data meanwhile enhancing generalization. In the long term, we aim to bridge the methodological gap between SFT and PO, striving for a more unified and flexible framework for adapting LLMs.

## 2 Related work

### 2.1 Data selection for SFT

SFT is a well-known fine-tuning technique that maximizes the likelihood of generating target tokens under the assumption that all tokens are informative. However, data quality has emerged as a critical bottleneck for this approach (Luo et al., 2024), with errors arising from various sources including human annotators, tool annotators, LLM hallucinations, and data processing inconsistencies (Luo et al., 2024).

LIMA (Zhou et al., 2023a), hypothesized that LLMs primarily learn the style of dataset responses, rather than updating their pre-trained knowledge toward specialized tasks, by showing that fine-tuning on a 10k carefully curated dataset, they can obtain better performance than fine-tuning on a larger dataset.

To address quality challenges, researchers have investigated the advantages of data quality over quantity, proposing selection algorithms based on quality and diversity metrics to filter misleading samples and improve instruction-following capabilities (Chen et al., 2023; Maharana et al., 2023; Lu et al., 2024; Wu et al., 2023; Xia et al., 2024). While effective at improving performance, these approaches suffer from a fundamental limitation: they operate at the sample level, discarding entire examples and thus reducing the overall data scale available for training. This creates an inevitable trade-off between quality and quantity that remains unresolved.

Several data quality metrics have been introduced, such as gradient matching (Zhou et al., 2023b), human feedback (Köpf et al., 2023) and influence function scores (Xia et al., 2024). Moreover, (Dai et al., 2025) demonstrated that naturally higher influence scores for certain tasks can introduce bias in data selection, and proposed normalizing influence scores across different tasks before iteratively selecting samples for under-represented skills. In (Luo et al., 2024), authors propose a two-stage noise-robust framework that performs noise detection using multiple expert systems and then relabels the downstream task data by finding similar examples from the clean set to provide context. In another approach, researchers showed that selecting training samples aligned with the model's existing knowledge can improve performance by generating multiple instruction-response pairs and choosing those with the highest probability according to the target model (Zhang et al., 2025).

Recent studies have explored various high-quality data selection algorithms for LLM fine-tuning, yet they predominantly overlook a crucial insight: even in noisy samples, some tokens still contain valuable information. By discarding entire samples, these methods inadvertently remove useful training signals. Furthermore, these approaches fail to utilize the rejected data as a learning signal.

## 2.2 LLM unlearning and PO

Several approaches have been proposed to remove specific information from LLM without complete retraining them from scratch, including data replacement and relabeling strategies (Eldan & Russinovich, 2023; jin et al., 2024), and knowledge editing techniques by predicting targeted parameter updates to change specific facts while preserving other knowledge (De Cao et al., 2021). Gradient ascent (GA) based methods are usually used for their simplicity, which maximize the negative log-likelihood of specific token sequences(Jang et al., 2023; Maini et al., 2024; Yao et al., 2024b; Tian et al., 2024; Cha et al., 2024; Yu et al., 2023). However, some of them lead to degradation in LLM's outputs globally and damage the overall integrity of LLMs when removing targeted knowledge (Yu et al., 2023; Wang et al., 2025a; Zhang et al., 2024; Lizzo & Heck, 2025)—called excessive unlearning, which some regularization techniques such as minimizing the KL-Div between the output distributions of the pre-trained and fine-tuned models (Yao et al., 2024a) is proposed to maintain performance on retain dataset. This introduce additional computational overhead and hyperparameter sensitivity. Researchers in (Wang et al., 2025b) introduced WGA, which applies confidence-based weights to mitigate the excessive unlearning on a controlled forgetting manner.

In the PO field, DPO has emerged as an alternative to PPO-based alignment methods. However, PPO has been successful for its sample efficiency compared to earlier policy gradient methods, it still suffers from explicitly modeling a reward model, and complex hyperparameter tuning (Schulman et al., 2017). To address these challenges and making it more robust and less computationally expensive, DPO formulates the alignment objective into a maximum likelihood formulation on a preference-paired data, trying to make preferred responses more likely and dispreferred responses less likely. There are extensive studies to address the limitations of DPO (Ethayarajh et al., 2024; Gheshlaghi Azar et al., 2024; Xu et al., 2024; Hong et al., 2024; Meng et al., 2024; Zeng et al., 2024), a new approach for preference-based unlearning was proposed by (Maini et al., 2024), which defines the forget set as the dispreferred responses, and the preferred response contains the refusal responses like "I do not know the answer". Inspired by this research, (Zhang et al., 2024) proposed a new variant of DPO, called negative preference optimization (NPO) that uses only negative responses, disregarding the positive ones. In the (Wang et al., 2025b) further proposed Token-level NPO (TNPO) and Weighted TNPO (WTNPO), applying unlearning at the individual token level for more precise control over knowledge removal, yet these methods were developed specifically for targeted forgetting rather than as a complement to learning during SFT.

# 3 Preliminary

In this section, we present the foundational background essential to our work. We start by introducing SFT for autoregressive language modeling, followed by discussing the data quality issues within SFT.

## 3.1 SFT

Autoregressive language modeling, known as sequential prediction of outputs conditioned on previous context, plays a dominant role in contemporary LLMs. After pre-training, SFT is typically adopted to further improve LLMs for specific tasks by optimizing on task-specific instruction-response pairs. Specifically, representing a training corpus as $D = \{(X_i, Y_i)\}_{i=1}^{N}$, including $N$ sequence sample pairs, each pair containing $X_i$ as an input prompt and $Y_i$ as a completion response. Each prompt $X_i$ is denoted as $X_i = \{x_{i,j}\}_{j=1}^{m_i}$ with $m_i$ indicating the sequence length of the $i$-th prompt. Similarly, each $i$-th completion response with sequence length of $n_i$ is denoted as $Y_i = \{y_{i,j}\}_{j=1}^{n_i}$. In an autoregressive manner, the model learns to estimate the probability distribution $P(y_{i,j}|X_i, y_{i,:j}; \theta)$ for each token $y_{i,j}$ in the response, conditioned on the entire prompt $X_i$ and all preceding generated tokens in the response $y_{i,:j} = \{y_{i,1}, y_{i,2}, \ldots, y_{i,j-1}\}$, where $\theta$ denotes the model parameters.

The standard cross-entropy objective is typically adopted for SFT, following the formulation of

$$\mathcal{L}(\theta) = \frac{1}{|\mathcal{I}|} \sum_{(i,j) \in \mathcal{I}} -\log P(y_{i,j}|X_i, y_{i,:j}; \theta), \tag{1}$$

where the index set is defined as:

$$\mathcal{I} := \{(i,j)|i \in \{1, 2, \ldots, N\}, j \in \{1, 2, \ldots, n_i\}\}, \tag{2}$$

and the per-token loss function is defined as:

$$\ell(y_{i,j}|x_{i,:j}; \theta) := -\log P(y_{i,j}|X_i, y_{i,:j}; \theta). \tag{3}$$

## 3.2 Data Quality of SFT

LLMs acquire diverse capabilities and knowledge representations through pretraining on extensive corpora. However, for utilizing them in specialized tasks, techniques such as SFT play a remarkable role in enhancing their performance by fine-tuning the LLM on the training corpus without any selection or discarding on the dataset's components (Pareja et al., 2025; Albalak et al., 2024).

However, collecting high-quality data, representing the required specific knowledge, is crucial to prevent inaccuracies and effectively align the LLM (Albalak et al., 2024). High-quality data collection can be challenging in practice due to several factors. Generally, task-specific data are often scarce, particularly in emerging domains. In addition, datasets are collected from various resources, often leading to inconsistent linguistic styles and quality, and errors due to the use of annotator tools, human manual annotating (Luo et al., 2024). Therefore, each of them can contribute noisy and misleading tokens into the dataset thus jeopardizing the optimization process, leading to poor generalization.

To mitigate the impacts of low-quality and misleading data/tokens, existing methods proposed various data selection methods to maintain beneficial and high-quality data for fine-tuning (Albalak et al., 2024). More specifically, existing methods address data filtering at the data level; however, token-level filtering seems to preserve dataset scale and fine-grained information much more.

Although progress has been made in previous studies, they discard the low-quality data during fine-tuning, which significantly reduces the original dataset scale and potentially limits the model generalization. This remains an open question: how to leverage the full training dataset at its original scale while improving model performance? Specifically, is it possible to not only learn from high-quality samples but also utilize misleading data/tokens to make improvements in model generalization without overfitting to noise, while maintaining the comprehensive scope of the original dataset?

## 4 Method

SFT is a well-established approach for aligning extensively knowledge-augmented pretrained LLMs with specialized tasks. As discussed in Section 3.2, practical datasets make it challenging for SFT to achieve high performance, as their collection process leads to a noisy dataset that jeopardizes the optimization process through misleading gradients. While many studies have attempted to address this issue by selecting high-quality subsets from SFT training data, these approaches sacrifice dataset scale instead of taking advantage from noisy tokens. This remained an open challenge to mitigate the effect of misleading tokens in the dataset, while preserving its scale. In this study, we propose a new approach for better LLM supervised fine-tuning, based on **forgetting** mechanism. Unlike traditional data selection approaches that treat all tokens uniformly and discard low-quality data, our method explicitly distinguishes between informative (positive) and uninformative or misleading (negative) tokens at a granular level. This token level approach preserves training data scale, while utilizing the tokens' training signals more effectively.

Specifically, actively forgetting negative tokens, rather than merely ignoring them, can significantly improve model performance by aligning better with target data, freeing up model capacity from undesired patterns, and preventing overfitting to noisy patterns. This insight is particularly valuable when working with practical datasets that inevitably include noisy tokens that should be forgotten to preserve the model's general capabilities. The overall pipeline is outlined in Algorithm 1. In the following parts, we introduce the components of our pipeline, including the data preprocessing and training objective function.

---

**Algorithm 1** Forgetting

---

**Require:** Base model $\theta$, dataset $\mathcal{D}$, proportion $\rho$, $t_{min}$, $t_{max}$
**Ensure:** Fine-tuned model $\theta^*$
 1: // Stage 1: Reference Model Fine-tuning
 2: $\theta' \leftarrow$ fine-tune $\theta$ on sampled subset $\mathcal{D}_{ref} \subset \mathcal{D}$
 3: // Stage 2: Token Quality Assessment
 4: $\mathcal{I} \leftarrow$ All token indices $(i, j)$ in $\mathcal{D}_{train}$
 5: **for** $(i, j) \in \mathcal{I}$ **do**
 6: $\quad Inf(y_{i,j}) \leftarrow \ell(y_{i,j}|x_{i,:j}; \theta') - \ell(y_{i,j}|x_{i,:j}; \theta)$
 7: $\quad \mathcal{Q}(y_{i,j}) \leftarrow -Inf(y_{i,j})$               ▷ Quality score
 8: **end for**
 9: // Stage 3: Token Selection
10: Sort tokens by $\mathcal{Q}(y_{i,j})$ to partition into positive and negative subsets
11: $\mathcal{P} \leftarrow \{(i, j) \in \mathcal{I} : \mathcal{Q}(y_{i,j}|x_{i,:j}; \theta, \theta') \geq \mathcal{F}_\mathcal{S}(1 - \rho)\}$     ▷ Positive tokens
12: $\mathcal{N} \leftarrow \mathcal{I} \setminus \mathcal{P}$                                  ▷ Negative tokens
13: // Stage 4: Training with Forgetting
14: **for** $step = 0$ to $total\_steps$ **do**
15: $\quad \lambda(step) \leftarrow (t_{max} - t_{min}) \cdot \frac{step}{total\_steps}$
16: $\quad \mathcal{L}_\mathcal{P} \leftarrow$ Mean weighted loss over positive tokens in $\mathcal{P}$
17: $\quad \mathcal{L}_\mathcal{N} \leftarrow$ Mean weighted loss over negative tokens in $\mathcal{N}$
18: $\quad \mathcal{L}(\theta) \leftarrow \mathcal{L}_\mathcal{P} - \lambda(step) \cdot \mathcal{L}_\mathcal{N}$
19: $\quad$ Update $\theta$ using optimizer step on $\mathcal{L}(\theta)$
20: **end for**
21: **return** $\theta$

---

### 4.1 Token quality assessment

To quantify token quality, we leverage the concept of influence functions (Koh & Liang, 2017), between the base and reference models. Given a base model with parameters $\theta$ and a reference model with parameters $\theta'$ (introduced in Section 5.1.2), we define the cross-model influence for token $y_{i,j}$ as follows.

$$Inf(y_{i,j}|x_{i,:j}; \theta, \theta') = \ell(y_{i,j}|x_{i,:j}; \theta') - \ell(y_{i,j}|x_{i,:j}; \theta). \tag{4}$$

The intuition is that tokens that become more predictable after initial training (resulting in loss reduction) represent patterns that the model has successfully learned and are likely to be informative.

The token quality score formulation is as follows:

$$\mathcal{Q}(y_{i,j}|x_{i,:j}; \theta, \theta') = -Inf(y_{i,j}|x_{i,:j}; \theta, \theta'). \tag{5}$$

A positive quality score indicates that the token became more predictable on the reference model (lower loss in $\theta'$ than in $\theta$), indicating that it represents a generalizable pattern. In contrast, a negative score suggests that the token might represent noise or misleading information.

### 4.2 Token selection

As a preprocessing step, we partition the tokens into positive and negative sets based on the quality scores. We first compute quality scores for all tokens in the training corpus, then sort them in descending order to form the set $\mathcal{S}$. Given a proportion hyperparameter $\rho \in (0, 1)$, we partition the tokens as follows:

$$\mathcal{P} = \{(i, j) \in \mathcal{I} : \mathcal{Q}(y_{i,j}|x_{i,:j}; \theta, \theta') \geq \mathcal{F}_\mathcal{S}(1 - \rho)\} \tag{6}$$
$$\mathcal{N} = \mathcal{I} \setminus \mathcal{P} \tag{7}$$

where $\mathcal{F}_\mathcal{S}(1-\rho)$ denotes the $(1-\rho)$-th percentile threshold in $\mathcal{S}$. The top $\rho$ proportion of tokens are considered as **positive** tokens form the $\mathcal{P}$ set, while the remaining tokens form the **negative** set $\mathcal{N}$. In practice, we found that setting $\rho$ in the range of 0.7 to 0.8 achieves best results in our experiments. Furthermore, our experiments reveal that partitioning tokens by a zero threshold score (i.e. $\mathcal{Q} > 0$ as positive tokens) negatively affects performance. This challenges the intuition that tokens with higher confidence improvement are informative and beneficial, while the others are harmful, introducing an open challenge for proposing more robust methods to identify high-quality tokens.

## 4.3 Training objective

While standard SFT algorithms maximize the likelihood over all tokens uniformly (potentially reinforcing noisy patterns that mislead optimization) and data selection methods discard the distinguished noisy data before training, our approach maintains the benefits of full-scale training while addressing quality concerns, which enables improvements in the model performance, by minimizing the likelihood of generating the noisy tokens and freeing model capacity from misleading noisy patterns. As mentioned in the Section 2.2, unlearning techniques proven to be effective to mitigate the influence of undesirable data while preserving the model utility. In our context, rather than **forgetting** some specified knowledge(e.g., copyrighted content), we forget misleading tokens through GA, effectively utilizing both positive and negative tokens. This approach enhances the model generalization while maintaining the original data scale with no information loss. We propose a training objective for our selective learning and **forgetting** as follows.

$$\mathcal{L}(\theta) = \frac{\sum_{(i,j)\in\mathcal{I}} y_{i,j} \cdot \mathbb{I}_{(i,j)\in\mathcal{P}} \cdot \ell(y_{i,j}|x_{i,:j};\theta)}{\sum_{(i,j)\in\mathcal{I}} y_{i,j} \cdot \mathbb{I}_{(i,j)\in\mathcal{P}}} - \lambda(step) \cdot \frac{\sum_{(i,j)\in\mathcal{I}} y_{i,j} \cdot \mathbb{I}_{(i,j)\in\mathcal{N}} \cdot \ell(y_{i,j}|x_{i,:j};\theta)}{\sum_{(i,j)\in\mathcal{I}} y_{i,j} \cdot \mathbb{I}_{(i,j)\in\mathcal{N}}}, \qquad (8)$$

where the first term represents the average weighted loss over positive tokens, and the second term represents the average weighted loss over negative tokens. We use $\lambda(step) = (t_{\max} - t_{\min}) \cdot \frac{\text{step}}{\text{total\_steps}}$ as an adaptive coefficient that scales linearly with training progress, ensuring an effective balancing of positive and negative gradients through the optimization process. Please refer to Appendix B.5 for more experiments on the $\lambda$ function selection.

In this training objective, optimization initially shares goals with generalization, but their objectives later diverge. The forgetting mechanism acts as a regularization technique that pulls optimization back for generalization when their goals conflict. By using the adaptive balancing coefficient, this enables to better capture the underlying preferred data distribution rather than overfitting to the noise or merely following the pattern of low-scale high-quality data.

However, our work differs from NPO (Zhang et al., 2024) and TNPO (Wang et al., 2025b) in problem setting and mechanism design. While NPO (Zhang et al., 2024) and TNPO (Wang et al., 2025b) address unlearning—removing predetermined unwanted knowledge (such as private data, copyrighted content) from trained models, our method focuses on SFT, where forgetting serves as a regularization term rather than the primary objective. We use token-level influence scores to automatically identify low-quality tokens within the training corpus to respect both the dataset quantity and quality. Then, apply forgetting to free model capacity from misleading noisy patterns, simultaneously learning positive tokens and forgetting negative ones. In contrast, NPO (Zhang et al., 2024) and TNPO (Wang et al., 2025b) operate on predefined forget sets where unlearning itself is the goal, not a regularization mechanism for improving generalization during task adaptation.

# 5 Experiments

## 5.1 Experimental setups

### 5.1.1 Datasets

**Training data.** We constructed our training corpus by randomly sampling from five datasets, Flan_v2 (Chung et al., 2024), Dolly (Conover et al., 2023), Open Assistant 1 (Köpf et al., 2023), Stanford Alpaca

Table 1: Dataset distribution comparison

| Dataset | 50k Sample | | 10k Sample | |
|---|---|---|---|---|
| | Samples | Percentage | Samples | Percentage |
| Dolly | 2,617 | 5.23% | 503 | 5.03% |
| Flan_v2 | 17,803 | 35.61% | 3,593 | 35.93% |
| Open Assistant 1 | 5,960 | 11.92% | 1,135 | 11.35% |
| Stanford Alpaca | 9,276 | 18.55% | 1,834 | 18.34% |
| WizardLM | 14,344 | 28.69% | 2,935 | 29.35% |

(Taori et al., 2023) and WizardLM (Xu et al., 2023). Please refer to Appendix A for more datasets details. The dataset distribution presented in detail in Table 1. This corpus provides a comprehensive coverage of domains and response styles, thereby enhancing the model's generalization capabilities (Wang et al., 2023).

**Evaluation benchmarks.** For the evaluation part, we have performed comprehensive evaluations on five diverse benchmark datasets. They are TruthfulQA (Lin et al., 2022) to evaluate the ability of LLM in providing truthful and accurate information, BoolQ (Clark et al., 2019) a binray question-answering dataset and evaluates LLM's ability in making precise boolean judgements, LogiQA (Liu et al., 2020) focused on logical reasoning, TydiQA (Clark et al., 2020) to evaluate the LLM on multilingual question-answering and ASDiv (Miao et al., 2020) to evaluate the LLM on math word problems. The benchmarks' attributes are presented in Table 2. The evaluation is processed on all benchmark samples, by using the lm-eval-hareness[1] repository.

### 5.1.2 Models

**Base models.** In this paper, we choose 3 open-source LLMs including LLaMA-3.2-1B, LLaMA-3.2-3B and LLaMA-3.1-8B (Grattafiori et al., 2024) in diverse complexity as our base models for fine-tuning.

**Reference models.** The reference models are obtained by fine-tuning the base models on a subset $\mathcal{D}_{\text{ref}} \subset \mathcal{D}$ with $\mathcal{D}_{\text{ref}} \cap \mathcal{D}_{\text{train}} = \emptyset$ where $\mathcal{D}_{\text{train}}$ is the training corpus and $\mathcal{D}$ is a combination of training datasets. The fine-tuned LLM will be used for calculating the influence scores. We also investigate the robustness of our approach when the reference dataset contains duplicate samples (see Appendix B.2).

**Baselines.** In this study, our baselines include the base model, the supervised fine-tuned version of the base model on the whole training dataset with full tokens, and the fine-tuned version of the base model on the preprocessed training dataset including only the top k% clean tokens.

### 5.1.3 Training configurations

For the reported results in Table 3, we employed model-specific hyperparameter pairs $(t_{\min}, t_{\max})$ as follows: $(10^{-5}, 0.25)$ for LLaMA-3.2-1B and $(10^{-4}, 0.25)$ for both LLaMA-3.2-3B and LLaMA-3.1-8B, for our adaptive balancing coefficient $\lambda(step)$. These values were determined through ablation studies optimizing for performance across our benchmark tasks. For fine-tuning the LLMs, we used LoRA (Hu et al., 2022) for its memory efficiency and stability during training. We set rank-size of 64, the scaling factor of 16 and dropout 0.1 for LoRA. We used the AdamW optimizer (Loshchilov & Hutter, 2019), with the overall batch size equal to 24 and the fine-tuning process is performed for 1 epoch with a learning rate $10^{-4}$ and a linear learning rate scheduler with 0.03 warm-up ratio. Moreover, we conducted our experiments on 4 NVIDIA L40S-48GB GPUs with Intel Xeon 6338 CPUs, running on Ubuntu 20.04.6 LTS. The systems utilize Transformers version 4.51.3 and CUDA version 12.5. Training time for 1B, 3B and 8B models approximately takes 2, 3, and 5 hours, respectively.

---

[1] https://github.com/EleutherAI/lm-evaluation-harness

Table 2: Evaluation datasets attributes

| Dataset | Focus Area | Data Size | Question Length |
|---------|------------|-----------|-----------------|
| TruthfulQA | Truthfulness | 817 | Medium |
| BoolQ | Boolean QA | 15,942 | Short |
| LogiQA | Logical reasoning | 8,678 | Medium |
| TydiQA | Multilingual QA | 204k | Varied |
| ASDiv | Math Word Problem Solving | 2,305 | Varied |

Table 3: Performance comparison of different methods across five different benchmarks using LLaMA-3.2-1B, LLaMA-3.2-3B and LLaMA-3.1-8B variants as our base models. We evaluate four approaches: Base (unmodified), Full Tokens (standard SFT), Ignoring, and our proposed **Forgetting**. The results show accuracy (%) for TruthfulQA, BoolQ, LogiQA, and ASDiv, and one-shot F1 score for TydiQA. Bold values demonstrate best performance on each benchmark. Results show mean values with standard deviations from 3 independent training runs. Our proposed Forgetting method achieves significant improvements across different benchmarks and model scales.

| Method | TruthfulQA | BoolQ | LogiQA | TydiQA | ASDiV | AVG |
|--------|-----------|-------|--------|--------|-------|-----|
| Base model: LLaMA-3.2-1B | | | | | | |
| Base | 37.83±0 | 63.80±0 | 22.17±0 | 14.36±0 | 0±0 | 27.63±0 |
| Full Tokens | 38.74±0.39 | 59.84±0.94 | 24.60±0.25 | 28.10±0.46 | 0.55±0.48 | 30.37±0.39 |
| Ignoring (seq-level) | 39.56±0.57 | 61.47±0.06 | 24.03±0.25 | 27.90±0.34 | 1.46±0.15 | 30.88±0.28 |
| Forgetting (seq-level) | 38.93±0.08 | 63.13±0.46 | 24.80±0.12 | 28.75±0.23 | 2.50±0.04 | 31.62±0.10 |
| Ignoring (token-level) | 42.40±0.13 | 60.21±1.66 | 24.34±0.31 | 33.87±0.64 | 0.91±0.2 | 32.35±0.46 |
| Forgetting (token-level) | **44.83±0.45** | **65.39±0.39** | **25.60±0.48** | **36.21±0.77** | **2.28±0.04** | **34.86±0.22** |
| Base model: LLaMA-3.2-3B | | | | | | |
| Base | 39.45±0 | 73.04±0 | 22.17±0 | 21.12±0 | 31.24±0 | 37.40±0 |
| Full Tokens | 42.95±0.47 | 72.54±0.59 | 25.51±0.21 | 44.04±0.27 | 49.46±0.14 | 46.90±0.16 |
| Ignoring (seq-level) | 40.58±0.54 | 72.93±0.28 | 24.36±0.47 | 44.82±1.03 | 49.11±0.29 | 46.36±0.41 |
| Forgetting (seq-level) | 40.95±0.30 | **77.80±0.38** | 25.27±0.23 | 47.52±0.45 | 49.83±0.93 | 48.27±0.06 |
| Ignoring (token-level) | 47.23±0.86 | 75.40±0.37 | 25.12±0.31 | 47.63±0.42 | 48.51±0.74 | 48.78±0.19 |
| Forgetting (token-level) | **50.32±0.96** | 76.66±0.07 | **27.09±0.37** | **56.36±0.06** | **50.47±0.3** | **52.18±0.12** |
| Base model: LLaMA-3.1-8B | | | | | | |
| Base | 45.08±0 | 82.15±0 | 26.51±0 | 46.67±0 | 12.93±0 | 42.67±0 |
| Full Tokens | 44.51±0.48 | 81.44±0.47 | 25.68±0.14 | 52.03±0.18 | 51.46±0.42 | 51.02±0.11 |
| Ignoring (seq-level) | 47.05±0.21 | 85.17±0.45 | 24.64±0.18 | 52.34±0.08 | 51.62±0.28 | 52.16±0.24 |
| Forgetting (seq-level) | 47.83±0.09 | **85.56±0.18** | 24.85±0.37 | 57.56±0.33 | 57.76±0.18 | 54.71±0.10 |
| Ignoring (token-level) | 52.38±0.22 | 82.76±0.07 | 25.53±0.11 | 56.66±0.06 | **57.95±0.35** | 55.06±0.16 |
| Forgetting (token-level) | **58.39±0.65** | 83.14±0.15 | **31.15±0.86** | **66.21±0.23** | 57.48±0.12 | **59.27±0.35** |

## 5.2 Empirical Results

We conducted comprehensive experiments to evaluate our forgetting approach against all baselines. Remarkably, our method outperformed all baselines in average performance. The forgetting method achieved superior results with $\rho$ in the range of 70% to 80%, while the ignoring has its best-case performance with $\rho$ in the range of 50% to 60% across all benchmarks. We demonstrate the results of our experiments utilizing three different variants of LLaMA in Table 3, comparing the method in their best-case performance, specifically, setting $\rho = 0.7$ for our forgetting approach and $\rho = 0.5$ for the ignoring approach. Notably, compared

to the standard SFT our method has achieved an average performance improvement of 4.49% on the 1B model, 5.28% on the 3B model and 8.25% on the 8B model. Furthermore, compared to ignoring baseline, our method has achieved performance improvement of 2.51% on the 1B model, 3.4% on the 3B model and 4.21% on the 8B model. Please see Appendix B.6 for a detailed analysis of computational overhead.

Additional experiments with LLaMA-2-13B (Touvron et al., 2023) confirms these forgetting mechanism's generalization capability in larger scales, with detailed results provided in Appendix B.1. To further validate the generalizability of our forgetting mechanism across different model architectures and benchmarks, we conducted additional experiments on Qwen2.5-3B (Qwen et al., 2025) and GPT-Neo-2.7B (Black et al., 2021) across four diverse benchmarks, Instruction-Following (Zhou et al., 2023c), ARC-Challenge (Clark et al., 2018), LAMBADA (Paperno et al., 2016) specially using OpenAI preprocessing (Radford et al., 2019) from the EleutherAI[2] repository, and Arithmetic (Brown et al., 2020). The results, presented in Appendix B.3, demonstrate the superiority of our forgetting mechanism. Notably, our forgetting method achieved a 5.33% improvement over the ignoring baseline on Qwen2.5-3B and a 3.56% improvement on GPT-Neo-2.7B, confirming that the benefits of our approach extend beyond the LLaMA family and linguistics task.

**Token-level vs. sequence-level granularity.** A key design choice in our approach is operating at the token level rather than the sequence level. This granular approach is motivated by the observation that individual sequences often contain a mixture of both informative and misleading tokens. Sequence-level selection would classify entire sequences as either positive or negative, potentially discarding valuable tokens within otherwise noisy sequences, or conversely, retaining harmful tokens within generally useful sequences. Token-level selection allows us to preserve beneficial information while selectively forgetting problematic content, maximizing the utility of our training data. The Table 3 shows a comparison of the different approaches.

Table 3 shows that token-level approaches consistently outperform their sequence-level counterparts across all model sizes. For example, with LLaMA-3.2-3B, token-level forgetting achieves 52.18% average performance compared to 48.27% for sequence-level forgetting. This superiority stems from token-level selection's ability to preserve useful information even in partially noisy sequences, while sequence-level selection discards entire sequences that may contain valuable tokens alongside problematic ones.

### 5.3   Ablation study

**Impact of $\rho$.** Our empirical evidence indicates that the forgetting approach demonstrates superior generalization capability when $\rho$ has a higher value, partitioning a larger subset of tokens as positive tokens and treating all remaining tokens as negative tokens (forget rate of $1 - \rho$). However, forgetting only a subset of the remaining tokens and discarding the others leads to suboptimal performance, indicating the effectiveness of forgetting all the $1 - \rho$ tokens as negative tokens. Figure 1(b) illustrates the average performance for different forget rates. Moreover, the choice of the hyperparameter $\rho$, directly affects the noise distribution in positive and negative sets. Higher value of $\rho$ can introduce noisy tokens to the positive set, while lower value of $\rho$ can add informative tokens to the negative set. Figure 1(a) shows the comparison between different values of $\rho$ for the forgetting and ignoring approaches. The average performance of the forgetting method has significantly decreased for the lowest value $\rho = 0.4$, due to the higher proportion of informative tokens in the negative set.

**Impact of $\lambda(step)$.** As explained in Section 4, effectively balancing the training and forgetting gradients is crucial for optimization stability. As related studies typically use a constant coefficient in the range (0,1) to reduce the learning rate of forgetting gradients. However, through empirical investigation, we observed that as training iterations progress, the learning rate reduction leads to the vanishing of the forgetting gradients. Thus, we used an adaptive function $\lambda(step)$, as a coefficient on forgetting loss term of our dual objective function, not only to balance the learning and forgetting gradients, but also to efficiently preserve the effects of forgetting gradients during fine-tuning. According to the dual objective function formula, ignoring approach is equivalent to forgetting with a balancing coefficient of zero. In a comparison of balancing coefficient strategies, we evaluated three approaches: static approaches with constant values zero (ignoring)

---

[2]https://huggingface.co/datasets/EleutherAI/lambada_openai

and 0.0001 (optimal value for static strategy), and a dynamic approach using the linear function $\lambda(step)$ with $t_{min} = 0.0001$ and $t_{max} = 0.25$. The corresponding average improvements are 48.78%, 49.59%, and 52.18%, respectively. These results demonstrate that adaptive adjustment via linear function significantly outperforms static coefficient assignment, highlighting the critical role of selecting an appropriate balancing coefficient strategy. By incorporating $\lambda(step)$, the forgetting learning rate decreases more gradually with a shallower slope. We investigated the impact of the adaptive parameter $\lambda(step)$ through a series of experiments.

**Hyperparameter sensitivity analysis.** To evaluate the robustness of our approach to hyperparameter choices, we conducted extensive experiments varying the key parameters $t_{min}$ and $t_{max}$ while keeping $\rho = 0.7$ fixed. As shown in Figure 1(a), our method demonstrates impressive robustness to $\rho$ values across a wide range. For practical selection of $\rho$, users can use the ratio of tokens with positive influence scores as an initial estimate—in our experiments, this ratio was 0.67, leading us to select $\rho = 0.7$ as optimal. Comprehensive results across different combinations of $t_{min}$ and $t_{max}$ values using LLaMA-3.2-3B are presented in Appendix B.4.

**Impact of forgetting.** As demonstrated in previous sections, the forgetting mechanism significantly improves the performance of fine-tuning with respect to that without forgetting and standard SFT. Specifically, when comparing the forgetting and ignoring approaches with the same selection ratio ($\rho = 0.7$), the forgetting method achieves an accuracy of 52.18%, outperforming the ignoring approach (48.39%). This performance gap indicates that the negative tokens set has a high noise ratio, reinforcing the impact of forgetting misleading tokens, leading to higher performance.

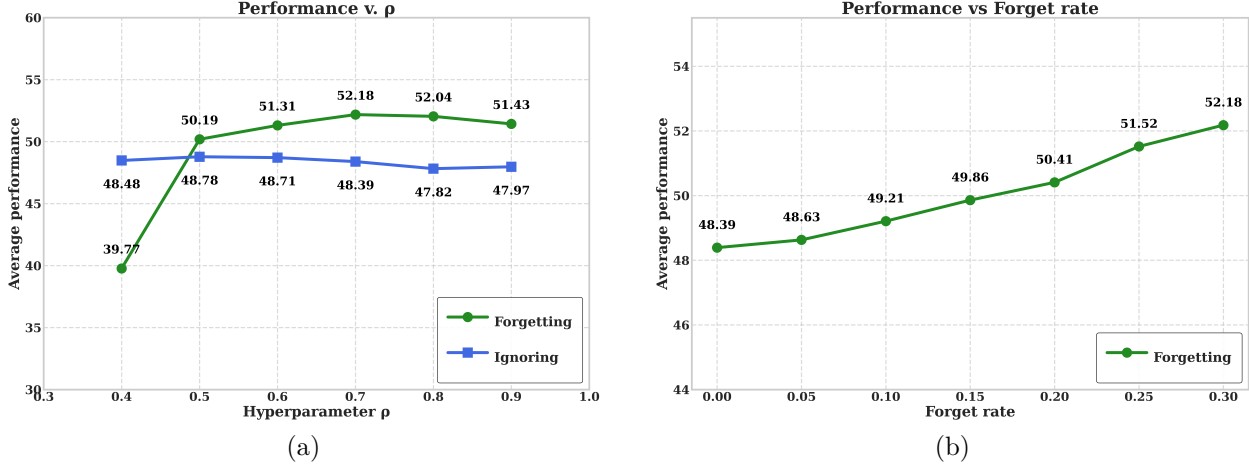

Figure 1: Performance analysis: (a) Average performance of forgetting versus ignoring methods across different $\rho$ values. (b) Average performance of the forgetting method with different forget rates.

## 6 Limitations

Despite our method's improvements, some limitations remain. The approach is sensitive to dataset size and noise ratio, leading to performance degradation for smaller negative token sets. However, it is worth noting that noise existence is common in real-world practical datasets. Additionally, computational budget restricted our experiments to models up to 13B parameters with limited-scale training data. The performance remains uncertain how well the mechanism would perform on larger-scale base models and datasets.

## 7 Conclusion

This paper aims to reduce the reliance of LLM fine-tuning on data quality, an important and on-going topic that has been receiving increasing attentions these days. Unlike previous works that primarily focus on improving data selection, we suggest that exploring new learning paradigms is equally crucial. Specifically, we propose a novel fine-tuning mechanism named forgetting, which explicitly enables the model to forget

misleading message carried by those filtered-out tokens. It mitigates the negative impact of noisy or misleading data while preserving the dataset scale, helping to improve generalization and overall performance. In the future, we will explore more formal and rigorous ways to defining and enhancing data quality, as well as extend the forgetting mechanism to other related areas within LLMs, such as pre-training, preference optimization, and inference.

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

## A Dataset Details

Table 4 provides comprehensive information about the datasets used to create training corpus, including their quality assessment, size, total length of samples, and source.

Table 4: Datasets attributes

| Dataset | Data Quality | Size | Length | Resource |
|---|---|---|---|---|
| Dolly | High | 15.01k | Varied | Human-annotated |
| Flan_v2 | High | 100k | Varied | Human-annotated |
| Open Assistant 1 | Moderate | 33.92k | Varied | Human-annotated |
| Stanford Alpaca | High | 52k | Varied | LLM-generated |
| WizardLM | High | 100k | Longer | LLM-generated |

## B Additional experimental results

### B.1 LLaMA-2-13B results

To further validate the robustness and scalability of our forgetting mechanism, we conducted additional experiments using LLaMA-2-13B as the base model. These results provide additional evidence that our approach consistently improves performance across different model architectures and scales, extending beyond the LLaMA-3.x series reported in the main paper.

The results in Table 5 demonstrate that our forgetting method maintains its effectiveness with larger models, achieving a 6.16% improvement over standard SFT and a 4.16% improvement over the ignoring baseline. This consistency across model scales (from 1B to 13B parameters) reinforces the generalizability of our approach and suggests that the forgetting mechanism provides fundamental benefits for supervised fine-tuning regardless of model size or architecture.

Table 5: Performance comparison of different methods across five benchmarks using LLaMA-2-13B as the base model. Results show accuracy (%) for TruthfulQA, BoolQ, LogiQA, and ASDiv, and one-shot F1 score for TydiQA. Bold values demonstrate best performance on each benchmark. Our proposed Forgetting method achieves significant improvements across different benchmarks, with an average improvement of 6.16% over standard SFT and 4.16% over the ignoring approach.

| Method | Dataset | | | | | |
|---|---|---|---|---|---|---|
| | TruthfulQA | BoolQ | LogiQA | TydiQA | ASDiV | AVG |
| Base model: LLaMA-2-13B | | | | | | |
| Base | 36.73 | 80.67 | 26.05 | 34.27 | 0.35 | 35.61 |
| Full Tokens (standard SFT) | 42.65 | 82.24 | 27.44 | 36.77 | 8.76 | 39.57 |
| Ignoring | 43.01 | **84.50** | 27.29 | 38.39 | 15.34 | 41.71 |
| Forgetting (Ours) | **52.82** | 84.13 | **27.95** | **48.71** | **17.80** | **46.28** |

### B.2 Impact of Reference Dataset Duplicates

We conducted additional experiments to investigate the robustness of our approach when the reference dataset contains duplicate samples. However our pipeline's preprocessing step removes duplicate samples

from the both training and references datasets, this analysis is important for understanding how data quality in the reference model training affects the overall forgetting mechanism performance.

Table 6 shows results using LLaMA-3.2-3B when the reference dataset includes duplicate samples. Interestingly, our forgetting method remains effective even under these suboptimal reference conditions, achieving a 4.93% improvement over standard SFT and a 2.05% improvement over the ignoring baseline. This demonstrates the robustness of our influence-based token quality assessment even when the reference model is trained on imperfect data, suggesting that our approach can handle practical scenarios where perfect data curation is not feasible.

Table 6: Performance comparison with duplicate samples in reference dataset using LLaMA-3.2-3B as base model. Results show mean values with standard deviations from 3 independent training runs. Our forgetting method maintains effectiveness even with imperfect reference data quality.

| Method | Dataset | | | | | |
|---|---|---|---|---|---|---|
| | TruthfulQA | BoolQ | LogiQA | TydiQA | ASDiV | AVG |
| Base model: LLaMA-3.2-3B (Reference with Duplicates) | | | | | | |
| Base | 39.45±0 | 73.04±0 | 22.17±0 | 21.12±0 | 31.24±0 | 37.40±0 |
| Full Tokens (standard SFT) | 42.95±0.47 | 72.54±0.59 | 25.51±0.21 | 44.04±0.27 | 49.46±0.14 | 46.90±0.16 |
| Ignoring | 49.91±0.39 | 75.60±0.86 | 24.99±0.35 | 48.61±0.20 | **49.81±0.01** | 49.78±0.22 |
| Forgetting (Ours) | **51.09±0.54** | **77.00±0.09** | **26.57±0.08** | **54.88±0.29** | 49.60±0.14 | **51.83±0.11** |

## B.3 Evaluation on Diverse Model Architectures

To demonstrate the broad applicability of our forgetting mechanism, we extended our evaluation to additional model architectures beyond the LLaMA family. Specifically, we conducted experiments on Qwen2.5-3B and GPT-Neo-2.7B, evaluating performance across four diverse benchmarks including Instruction-Following (IFEval), ARC-Challenge, LAMBADA, and Arithmetic. The characteristics of these evaluation benchmarks are detailed in Table 7.

For these experiments, we maintained our LLaMA-3.2-3B experimental setup and hyperparameters, as described in Section 5.1.3. The results presented in Table 8 show that our forgetting method consistently outperforms both standard SFT (full tokens) and the ignoring baseline across both model architectures. On Qwen2.5-3B, our method achieves an average performance of 59.01%, representing a 16.49% improvement over standard SFT and a 5.33% improvement over the ignoring approach. Similarly, on GPT-Neo-2.7B, our forgetting mechanism attains 28.15% average performance, demonstrating a 4.37% improvement over standard SFT and a 3.56% improvement over ignoring. These results confirm that the effectiveness of our forgetting mechanism generalizes well across diverse model architectures and evaluation tasks, validating its broad applicability for improving SFT of large language models.

Table 7: Characteristics of diverse evaluation benchmarks

| Dataset | Focus Area | Data Size | Question Length |
|---|---|---|---|
| IFEval | Instruction Following | 541 | Varied |
| ARC-Challenge | Scientific Reasoning | 1,172 | Medium |
| LAMBADA | Language Modeling | 5,153 | Short |
| Arithmetic | Mathematical Computation | Varied | Short |

## B.4 Hyperparameter sensitivity analysis

Table 9 presents comprehensive results across different combinations of $t_{\min}$ and $t_{\max}$ values using LLaMA-3.2-3B. The results demonstrate remarkable stability, with performance variations remaining small across different hyperparameter settings (standard deviation $< 0.5\%$ across configurations). This robustness ensures

Table 8: Performance comparison across diverse model architectures and benchmarks. Results show accuracy (%) for all benchmarks. Bold values indicate best performance. Our forgetting method demonstrates consistent improvements across different model families (Qwen and GPT-Neo), validating its broad applicability beyond the LLaMA architecture family.

| Model | Method | IFEval | ARC-Challenge | LAMBADA | Arithmetic | AVG |
|---|---|---|---|---|---|---|
| | Base model: Qwen2.5-3B | | | | | |
| Qwen2.5-3B | Base | 23.13 | 44.70 | 66.72 | 13.16 | 36.93 |
| | Full Tokens | 19.59 | 43.66 | 68.64 | 38.19 | 42.52 |
| | Ignoring | 19.22 | **45.63** | 68.82 | 81.03 | 53.68 |
| | Forgetting (Ours) | **33.49** | 45.39 | **69.76** | **87.40** | **59.01** |
| | Base model: GPT-Neo-2.7B | | | | | |
| GPT-Neo-2.7B | Base | 1.90 | 27.48 | 61.74 | 0.43 | 22.89 |
| | Full Tokens | 1.49 | 29.72 | 63.05 | 0.84 | 23.78 |
| | Ignoring | 0.19 | 30.55 | 66.15 | 1.46 | 24.59 |
| | Forgetting (Ours) | **11.66** | **31.63** | **67.29** | **2.03** | **28.15** |

that our method maintains superiority over baselines without requiring extensive hyperparameter tuning. The stability is partly attributed to the inherent robustness of large language models and their extensive pre-trained knowledge, which provides a strong foundation that is resilient to moderate changes in fine-tuning parameters.

Table 9: Hyperparameter sensitivity analysis for $t_{\min}$ and $t_{\max}$ using LLaMA-3.2-3B with fixed $\rho = 0.7$. Results demonstrate robustness across different parameter combinations.

| $t_{\min}$ | $t_{\max}$ | TruthfulQA | BoolQ | LogiQA | TydiQA | ASDiV | AVG |
|---|---|---|---|---|---|---|---|
| 0.00001 | 0.45 | 52.75 | 74.38 | 25.89 | 54.27 | 48.10 | 51.08 |
| 0.00001 | 0.35 | 51.55 | 75.11 | 26.15 | 56.74 | 48.42 | 51.59 |
| 0.00001 | 0.25 | 50.93 | 76.58 | 25.99 | 56.13 | 50.26 | 51.98 |
| 0.00001 | 0.15 | 50.17 | 75.45 | 26.19 | 54.37 | 50.67 | 51.37 |
| 0.0001 | 0.45 | 50.90 | 77.56 | 25.83 | 54.33 | 48.90 | 51.50 |
| 0.0001 | 0.35 | 51.20 | 75.67 | 26.65 | 57.21 | 48.78 | 51.90 |
| 0.0001 | 0.25 | 50.32 | 76.64 | 27.09 | 56.36 | 50.47 | **52.18** |
| 0.0001 | 0.15 | 50.09 | 74.79 | 25.27 | 55.21 | 51.82 | 51.44 |
| 0.001 | 0.15 | 49.05 | 76.03 | 26.36 | 54.85 | 51.49 | 51.56 |
| 0.001 | 0.25 | 48.96 | 76.50 | 28.68 | 56.35 | 49.66 | 52.03 |
| 0.001 | 0.35 | 51.25 | 74.41 | 26.51 | 56.58 | 50.05 | 51.76 |
| 0.001 | 0.45 | 50.69 | 74.50 | 25.98 | 56.97 | 48.46 | 51.32 |
| 0.01 | 0.15 | 50.46 | 75.24 | 26.12 | 54.17 | 50.95 | 51.39 |
| 0.01 | 0.25 | 51.02 | 76.28 | 27.75 | 55.48 | 49.93 | 52.09 |
| 0.01 | 0.35 | 52.78 | 74.44 | 25.58 | 55.92 | 48.30 | 51.40 |
| 0.01 | 0.45 | 50.09 | 74.87 | 27.60 | 54.69 | 48.68 | 51.19 |

### B.5   $\lambda(step)$ **vs Constant** $\lambda$

In this section, we compare our adaptive function $\lambda(step)$ against using a constant value for $\lambda$. To ensure a fair comparison, we conducted extensive experiments on LLaMa-3.2-3B, evaluating a wide range of constant values. Table 10 presents the results across different constant settings, demonstrating that even the best-performing constant value is outperformed by our adaptive $\lambda(step)$ approach.

Table 10: $\lambda(step)$ selection experiments on LLaMA-3.2-3B with fixed $\rho = 0.7$.

| $\lambda$ | TruthfulQA | BoolQ | LogiQA | TydiQA | ASDiV | AVG |
|---|---|---|---|---|---|---|
| constant value | 47.85 | 76.53 | 25.19 | 50.41 | 49.27 | 49.85 |
| $(t_{\max} - t_{\min}) \cdot \frac{step}{total\_steps}$ (linear) | **50.32** | **76.64** | **27.09** | **56.36** | **50.47** | **52.18** |

### B.6   **Computational Cost**

As shown in the Table 11, we conducted a detailed analysis of computational cost. Given the consistent and significant performance improvements across different scales, this computational overhead is reasonable and justified. The preprocessing phase (warmup + influence computation) is a one-time cost that can be ignored if multiple training runs are conducted.

Table 11: Training time comparison of different strategies for 3B and 8B models.

| Component | 3B Model | 8B Model |
|---|---|---|
| Preprocessing Phase | | |
| Warmup Training | 36 min | 52 min |
| Influence Score Computation | 47 min | 73 min |
| Training Phase | | |
| Full Token | 205 min | 347 min |
| Ignoring | 166 min | 282 min |
| Forgetting | 187 min | 311 min |
| Total Time | | |
| Full Token (training only) | 205 min | 347 min |
| Ignoring (warmup + score + training) | 249 min | 407 min |
| Forgetting (warmup + score + training) | 270 min | 436 min |
| Overhead | | |
| Ignoring vs Full Token | +21.46% | +17.29% |
| Forgetting vs Full Token | +31.71% | +25.65% |
| Forgetting vs Ignoring | +8.43% | +7.12% |

