# OpenReview forum: "Forgetting: A New Mechanism Towards Better Large Language Model Fine-tuning"
_TMLR — Accepted by TMLR_

### Review · Reviewer_ggfb · 2025-12-08

**Summary Of Contributions:**

The paper proposes a token quality assessment within the SFT framework in order to better utilize the tokens' training signals. To do so, the method relies on the cross-model influence metric to come up with a token quality score. The tokens in the corpus are then sorted according to this quality score and positive/negative tokens are determined by slicing this ranking according to percentiles. Negative tokens are then forgotten during the training process using gradient ascent.

**Audience:**

Yes

**Audience Explanation:**

Yes, this work is of interest to folks in the community working on post training optimization such as SFT.

**Broader Impact Concerns:**

No such concerns from my side.

**Claims And Evidence:**

Yes

**Claims Explanation:**

The proposed method is sound, the equations are meaningful and appear to be right. The empirical results support the applicability of the method.

**Requested Changes:**

No changes requested - good paper.

---

> ### Author Response · Authors · 2026-01-25
> **Response to Reviewer ggfb**
>
> We sincerely thank the reviewer for taking the time to carefully read and review our paper. We appreciate your thoughtful feedback and positive assessment of our work.
>
> Sincerely,
>
> The Authors of Submission 6522

---

### Review · Reviewer_egFx · 2025-12-26

**Summary Of Contributions:**

This paper proposes Forgetting, a novel LLM fine-tuning method that focuses on lower-quality data. Unlike prior work that performs sequence-level filtering, the paper introduces a token-level filtering approach that classifies each token as positive or negative. Moreover, building on an intuition similar to that of preference optimization methods, the negative tokens are explicitly forgotten (i.e., unlearned).

Empirical evaluations on a range of models with sizes up to 13B parameters demonstrate a significant improvement in model performance with the proposed method. The ablation study further shows that both token filtering and unlearning are effective.

**Audience:**

Yes

**Audience Explanation:**

SFT is an important problem for the community and would have quite large audience.

**Broader Impact Concerns:**

Not any

**Claims And Evidence:**

Yes

**Claims Explanation:**

The proposed method is simple, intuitive, and effective, and it is backed by sound empirical evaluation. The experiments cover models of different sizes and multiple datasets.

For the two fundamental contributions—token-level filtering and unlearning—the ablation study demonstrates that both components are effective, and that combining them achieves the best performance. These results support the theoretical claims made in the earlier sections, namely that finer-grained filtering is beneficial and that negative signals should not be ignored.

**Requested Changes:**

From my perspective, no major changes are required. I only have a few minor recommendations:
- Providing more details on how the reference model is trained would further strengthen the claims. For example, what proportion of the dataset is used to train the reference model, and does this incur significant additional computational overhead?
- It would be helpful to include more theoretical or intuitive explanations for the choice of the adaptive coefficient. In particular, this could better support the claim that the strength of unlearning should be gradually increased over time.
- Additionally, although it is not directly related to the paper’s main claims, it would be interesting to analyze which tokens tend to be classified as positive or negative, and whether the filtering exhibits any bias toward particular types of tokens or token distributions.

---

> ### Author Response · Authors · 2025-12-26
> **Response to Reviewer egFx**
>
> We sincerely thank the reviewer for the thoughtful evaluation and constructive feedback. Below we address each of your recommendations.
>
> > Q1. More details on reference model training
>
> **A1.** Thank you for this suggestion. We provide the following clarifications:
>
> **Dataset proportion:** The reference model is trained on a 10k sample subset, while the main training corpus contains 50k samples. Standard SFT (Full Token baseline) uses the full 60k samples (10k + 50k). This ensures a fair comparison as both approaches utilize the same total amount of data.
>
> **Computational overhead:** We conducted detailed timing analysis as follows:
>
> | Component | 3B Model | 8B Model |
> |-----------|----------|----------|
> | **Preprocessing Phase** | | |
> | Warmup Training | 36 min | 52 min |
> | Influence Score Computation | 47 min | 73 min |
> | **Training Phase** | | |
> | Full Token | 205 min | 347 min |
> | Ignoring | 166 min | 282 min |
> | Forgetting | 187 min | 311 min |
> | **Total Time** | | |
> | Full Token (training only) | 205 min | 347 min |
> | Ignoring (warmup + score + training) | 249 min | 407 min |
> | Forgetting (warmup + score + training) | 270 min | 436 min |
> | **Overhead** | | |
> | Ignoring vs Full Token | +21.46% | +17.29% |
> | Forgetting vs Full Token | +31.71% | +25.65% |
> | Forgetting vs Ignoring | +8.43% | +7.12% |
>
> Given the consistent and significant performance improvements across different scales, this computational overhead is reasonable and justified. Notably, the preprocessing phase (warmup + influence computation) is a one-time cost that can be amortized if multiple training runs are conducted.
>
> > Q2. intuitive explanation for the adaptive coefficient λ(step)
>
> **A2.** We appreciate this suggestion and provide the following intuition.
>
> In early training stages, the model is still learning fundamental patterns from positive tokens. Applying strong forgetting gradients prematurely can destabilize optimization before the model has established a solid foundation of correct knowledge. A small initial λ (e.g., 10⁻⁴) allows the learning signal to dominate. As training progresses, the model begins to overfit to noise in the training data. At this stage, stronger forgetting gradients (larger λ) serve as an effective regularizer, actively pushing the model away from memorizing misleading patterns and establishing clearer knowledge boundaries.
>
> Table 10 in Appendix B.5 demonstrates that our linear schedule (λ(step) = (t_max - t_min) · step/total_steps) achieves 52.18% average performance compared to 49.85% for the best constant λ, validating this adaptive design.

---

### Review · Reviewer_sGcN · 2026-01-22

**Summary Of Contributions:**

The paper proposes a token-level “forgetting” mechanism in SFT. It splits tokens into positive vs negative based on the loss difference between a base and reference model. It trains the model with a dual objective: standard likelihood training on positive tokens and gradient-ascent-style “forgetting” (rather than discarding) on negative tokens. The method yields consistent average improvements over standard SFT and “ignoring” baselines across multiple model sizes and architectures.

strengths
- The paper proposes a token-level “forgetting”, and compares token-level vs sequence-level and shows token-level doing better
- The paper proposes to “forget” low-quality tokens instead of just discarding them
- Experiments show consistent performance gains across different model scales on diverse benchmarks

weaknesses
- The "forgetting" mechanism is essentially gradient ascent on negative samples, similar to DPO/NPO negative component.
- The method requires training a reference model first.
- Missing qualitative analysis of what tokens are being classified as negative.
- Missing comparison with recent advances.

**Audience:**

Yes

**Audience Explanation:**

The data quality and selection is a practically important problem in SFT, which is highly relevant to ML community.

**Broader Impact Concerns:**

The reference model could be biased without carefully safety/regression testing.

**Claims And Evidence:**

No

**Claims Explanation:**

The paper provides emprical evidence that adding the forgetting term improves average benchmark scores compared to standard SFT and Ignoring mechanisms.

However, the concepts of "negative tokens lack essential semantics or are misleading", is not directly supported. It is measured by the cross-model loss change. The metric introduces several possible factors, such as reference model fine-tuning may introduce distribution shift or catastrophic forgetting on rare tokens. The paper would benefit from qualitative and quantitative analyses of what kinds of tokens are classified negative.

The claims about “shapes a knowledge boundary” and improves “diversity” are also not directly supported.

**Requested Changes:**

- Add comparison with stronger recent advances for noisy/robust SFT or data selection under matched compute and data, such as D^2pruning, LESS.
- Report the cost of the method such as reference training + token scoring + final training (time/GPUs), and discuss scalability for larger models/datasets.
- Provide qualitative and quantitative analyses of what kinds of tokens are classified negative, such as token type, position, dataset sources, etc., and discuss whether the “negative” tokens are really correspond to actual annotation noise, factual errors, or low-quality generations.
- Discuss how reference model quality affects results.
- Weaken or remove the claim about “shapes a knowledge boundary” and improves “diversity”, unless providing more evidences.

---

> ### Author Response · Authors · 2026-01-22
> **Response to Reviewer sGcN**
>
> Thank you for your thorough review and constructive feedback. Below we address each of your concerns directly.
>
>  > ### **Similarity to DPO/NPO negative component**
>
> We acknowledge the mathematical similarity but emphasize that we have already clarified the fundamental differences in Section 4.3 of our paper:
>
> *"However, our work differs from NPO (Zhang et al., 2024) and TNPO (Wang et al., 2025) in problem setting and mechanism design. While NPO (Zhang et al., 2024) and TNPO (Wang et al., 2025) address unlearning—removing predetermined unwanted knowledge (such as private data, copyrighted content) from trained models, our method focuses on SFT, where forgetting serves as a regularization term rather than the primary objective. We use token-level influence scores to automatically identify low-quality tokens within the training corpus to respect both the dataset quantity and quality. Then, apply forgetting to establish clearer knowledge boundaries, simultaneously learning positive tokens and forgetting negative ones. In contrast, NPO (Zhang et al., 2024) and TNPO (Wang et al., 2025) operate on predefined forget sets where unlearning itself is the goal, not a regularization mechanism for improving generalization during task adaptation."*
>
> ----
>
> > ### **Reference model requirement & Computational cost**
>
> We have previously addressed this concern in response to another reviewer. Here is our detailed analysis:
>
> **Data usage fairness:**
> The reference model is trained on a 10k sample subset, while the main training corpus contains 50k samples. Standard SFT (Full Token baseline) uses the full 60k samples (10k + 50k). This ensures a fair comparison as both approaches utilize the same total amount of data.
>
> **Computational overhead:**
>
> | Component | 3B Model | 8B Model |
> |-----------|----------|----------|
> | **Preprocessing Phase** | | |
> | Warmup Training | 36 min | 52 min |
> | Influence Score Computation | 47 min | 73 min |
> | **Training Phase** | | |
> | Full Token | 205 min | 347 min |
> | Ignoring | 166 min | 282 min |
> | Forgetting | 187 min | 311 min |
> | **Total Time** | | |
> | Full Token (training only) | 205 min | 347 min |
> | Ignoring (warmup + score + training) | 249 min | 407 min |
> | Forgetting (warmup + score + training) | 270 min | 436 min |
> | **Overhead** | | |
> | Ignoring vs Full Token | +21.46% | +17.29% |
> | Forgetting vs Full Token | +31.71% | +25.65% |
> | Forgetting vs Ignoring | +8.43% | +7.12% |
>
> Full Token baseline trains on all 60k samples directly. Our method first trains a reference model on 10k samples, computes influence scores, then trains on the remaining 50k samples with forgetting. **The total data volume remains constant** and we simply allocate it differently. Moreover, the preprocessing phase (warmup + influence computation) is a one-time cost that can be amortized if multiple training runs are conducted.
>
> **Scalability discussion:**
> We have comprehensively addressed scalability concerns in Section 6 (Limitations) of our paper, where we discuss the computational budget constraints and note that performance on larger-scale models and datasets requires further investigation.
>
> ---
>
> > ### **qualitative analysis of negative tokens**
>
> We appreciate this suggestion. However, it is important to emphasize that **our primary objective is to demonstrate the effectiveness of forgetting mechanisms in enhancing generalization performance, not to definitively categorize "good" and "bad" tokens**. Our contribution is methodological; we show that using influence-based token selection combined with forgetting yields consistent improvements across diverse settings. Token-level linguistic analysis is beyond the scope of our current work which focuses on demonstrating the performance benefits of the forgetting mechanism.
>
> ---
>
>
> > ### **Reference model quality effects**
>
> We have addressed this in **Appendix B.2: Impact of Reference Dataset Duplicates**, where we show that even when the reference dataset contains duplicate samples, our forgetting method maintains strong performance.
>
> ---
>
> > ### **Claims about "knowledge boundary" and "diversity"**
>
> We will reframe "knowledge boundary" as *"provides clearer training signals by distinguishing between tokens to learn and tokens to forget"* in the revised manuscript. Regarding "diversity," we will remove this claim entirely.
>
> -----
>
> Thank you for your valuable feedback.
>
> Sincerely,
> The Authors of Submission 6522

---

> > ### Comment · Reviewer_sGcN · 2026-01-26
> >
> > Thank you for the detailed response especially regarding the computational overhead and the adjustments to the claims about "diversity" and "knowledge boundaries”.
> >
> > ## Similarity to DPO/NPO negative component
> > The “problem setting differences” (SFT vs. unlearning) mentioned in Sec4.3 is more to the application context, not fundamental novelty. Could you provide a mathematical comparison demonstrating the differences? Are they both gradient ascent on negative samples weighted by an adaptive coefficient? I would appreciate a clearer claim & comparison to help audience better understand your novelty. Generally I acknowledge your contribution in "adapting forgetting mechanisms to SFT with influence-based token selection."
> >
> > Moreover, since your paper proposes a specific metric for selection, I would recommend you to empirically compare and demonstrate your influence-based selection is superior to established data selection baselines.
> >
> > ## Qualitative analysis of negative tokens
> > This analysis would be help convince your claims that “the negative tokens are "misleading" or "lack essential semantics" in the abstract. Also it will help readers understand whether the method is truly targeting noise/misinformation, or merely suppressing hard/rare tokens etc.
> >
> > ## Reference model quality effects
> > The duplicate-reference experiment is a useful testing, but it may not fully address the broader concern on how sensitive are results to the reference model’s quality. Some stronger evidences could be ablations on reference set size, reference data domain distributions. In table3, you mentioned “Results show mean values with standard deviations from 3 independent training runs.” Did you conduct random split 3 times and trained 3 separated reference models across 3 runs?

---

> ### Author Response · Authors · 2026-01-29
> **Response to Reviewer sGcN**
>
> Thank you for your constructive feedback. Below, we address each of your concerns.
>
> > **Similarity to DPO/NPO negative component**
>
> Here, we provide a detailed analysis to clearly demonstrate our novelty.
>
> DPO's objective can be written as:
>
> $$L_{DPO}(\theta) = -\mathbb{E} \left[\log \sigma\left(\beta \log \frac{\pi_{\theta}(y_w|x)}{\pi_{ref}(y_w|x)} - \beta \log \frac{\pi_{\theta}(y_l|x)}{\pi_{ref}(y_l|x)}\right)\right]$$
>
> where $y_w$ represents the preferred response, $y_l$ represents the dispreferred response, $\beta$ is the temperature parameter, and $\pi_{\text{ref}}$ is the reference policy which remains fixed. This formulation implicitly combines a positive component that increases the likelihood of $y_w$, a negative component that decreases the likelihood of $y_l$, and regularization through KL divergence constraints via the reference model.
>
> NPO's objective can be written as:
>
> $$L_{NPO}(\theta) = -\mathbb{E} \left[\log \frac{\pi_{\theta}(y_l|x)}{\pi_{ref}(y_l|x)}\right]$$
>
> This approach focuses only on negative samples (dispreferred responses) and performs gradient ascent on these negative samples with implicit reference model regularization.
>
> The problem setting differs fundamentally between DPO/NPO and our approach. DPO and NPO address preference optimization where predefined preference pairs are available, while our forgetting mechanism operates in the supervised fine-tuning setting, where we have a single corpus with mixed quality, through automatically identifying negative tokens via influence-based token quality scores, eliminating the need for manual preference annotation.
>
> In DPO/NPO, the reference model leverages in regularization through a KL constraint, in the form of $\mathbb{KL}(\pi_\theta || \pi_{\text{ref}})$ that appears directly in the optimization objective, while our method does not use KL regularization in the training objective.
> The adaptive weighting strategy differs as well. DPO uses a fixed $\beta$ parameter for controlling the deviation from the base reference policy, while we employ a time-dependent $\lambda(t)$ for our forgetting regularization term.
>
> > **Reference model quality effects**
>
> Thank you for your insightful comment. Here, we provide the results of creating the reference model on the 5k dataset, which we had already conducted. As shown in the table below, our reference model's quality stays robust in different situations.
>
> | Model      | TruthfulQA | BoolQ | LogiQA | TydiQA | ASDiv | AVG    |
> |------------|------------|-------|--------|--------|-------|--------|
> | Ignoring   | 46.73      | 75.97 | 25.23  | 49.2   | **49.22** | 49.27  |
> | Forgetting | **49.74**      | **77.24** | **26.55**  | **53.87**  | 48.49 | **51.18** |
>
> Moreover, about the “Results show mean values with standard deviations from 3 independent training runs,” we meant that we conducted our experiment runs on 3 different seed values for all of the models.

---

### Review · Reviewer_NBXC · 2026-01-29

**Summary Of Contributions:**

The paper introduces a "forgetting" mechanism for LLM SFT, classifying tokens as positive (informative) or negative (misleading) via influence functions from a reference model. Positive tokens use standard MLE; negative ones apply gradient ascent with adaptive balancing. This preserves data scale, mitigates noise, and improves generalization, shown via gains (2.51–8.25%) on benchmarks with LLaMA, Qwen, and GPT-Neo models.

### Key Strengths
- Token-level forgetting preserves scale while addressing noise, outperforming sample-level filtering.
- Strong experiments across models, tasks, and ablations; links SFT to PO for unified adaptation.

### Key Weaknesses
 None.

**Audience:**

Yes

**Audience Explanation:**

Researchers working on LLM unlearning would be interested in these findings.

**Broader Impact Concerns:**

None.

**Claims And Evidence:**

Yes

**Claims Explanation:**

Performance claims backed by averaged results on 5+ benchmarks, ablations on ρ/λ/granularity, and appendices for other models/tasks. Methods clear with equations/algorithm; "knowledge boundaries" inferred from gains, not directly probed.

**Requested Changes:**

- **Clarify examples (strengthening):** Add positive/negative token examples with scores/context.
- **Compare unlearning baselines (critical):** Benchmark vs. NPO/TNPO in SFT.
- **Analyze boundaries (strengthening):** Include output probes pre/post-forgetting.
- **Scale tests (strengthening):** Discuss/add larger models/datasets.
- **Fix issues (critical):** Correct citations/typos; define w_{i,j}.
- **Costs (strengthening):** Table for overhead vs. SFT.
- **Clarify 2.1 (critical):** Define "noisy samples" (e.g., annotation errors); add: "Hence, we adopt token-level filtering to retain useful signals in noisy samples."
- **Alternative metrics (strengthening):** Ablate token quality with gradients/loss/confidence.

---

> ### Author Response · Authors · 2026-01-29
>
> We appreciate your insightful comments. Here we provide the explanations.
>
> > **Compare unlearning baselines (critical): Benchmark vs. NPO/TNPO in SFT.**
>
>  NPO/TNPO are variants of DPO for removing some specific knowledge from the LLM. These methods require a predefined forgetting dataset (containing knowledge to remove) and a retaining dataset (containing knowledge to preserve), which fundamentally differs from our SFT setting. So, to our knowledge, making them less suitable as baselines.
>
> > **Scale tests (strengthening): Discuss/add larger models/datasets.**
>
> As discussed in Section 6 (Limitations), our computational budget restricted experiments to models up to 13B parameters. We have included results for LLaMA-2-13B in Appendix B.1, demonstrating consistent improvements. Extending to larger model/dataset scales could remain future work.
>
> > **Fix issues (critical): Correct citations/typos; define w_{i,j}.**
>
> We thank the reviewer for this observation. The corrected formulation is represented below as standard cross-entropy
> $$L(\theta) = \frac{1}{|I|} \sum_{(i,j) \in I} -\log P(y_{i,j} | X_i, y_{i,:j}; \theta).$$
>
> > **Costs (strengthening): Table for overhead vs. SFT.**
>
> We conducted a detailed timing analysis as follows:
>
> We appreciate your insightful comments. Here we provide the explanations.
>
> > **Compare unlearning baselines (critical): Benchmark vs. NPO/TNPO in SFT.**
>
>  NPO/TNPO are variants of DPO for removing some specific knowledge from the LLM. These methods require a predefined forgetting dataset (containing knowledge to remove) and a retaining dataset (containing knowledge to preserve), which fundamentally differs from our SFT setting. So, to our knowledge, making them less suitable as baselines.
>
> > **Scale tests (strengthening): Discuss/add larger models/datasets.**
>
> As discussed in Section 6 (Limitations), our computational budget restricted experiments to models up to 13B parameters. We have included results for LLaMA-2-13B in Appendix B.1, demonstrating consistent improvements. Extending to larger model/dataset scales could remain future work.
>
> > **Fix issues (critical): Correct citations/typos; define w_{i,j}.**
>
> We thank the reviewer for this observation. The corrected formulation is represented below as standard cross-entropy
> $$L(\theta) = \frac{1}{|I|} \sum_{(i,j) \in I} -\log P(y_{i,j} | X_i, y_{i,:j}; \theta).$$
>
> > **Costs (strengthening): Table for overhead vs. SFT.**
>
> We conducted a detailed timing analysis as follows:
>
> | Component | 3B Model | 8B Model |
> |-----------|----------|----------|
> | **Preprocessing Phase** | | |
> | Warmup Training | 36 min | 52 min |
> | Influence Score Computation | 47 min | 73 min |
> | **Training Phase** | | |
> | Full Token | 205 min | 347 min |
> | Ignoring | 166 min | 282 min |
> | Forgetting | 187 min | 311 min |
> | **Total Time** | | |
> | Full Token (training only) | 205 min | 347 min |
> | Ignoring (warmup + score + training) | 249 min | 407 min |
> | Forgetting (warmup + score + training) | 270 min | 436 min |
> | **Overhead** | | |
> | Ignoring vs Full Token | +21.46% | +17.29% |
> | Forgetting vs Full Token | +31.71% | +25.65% |
> | Forgetting vs Ignoring | +8.43% | +7.12% |
>
> Given the consistent and significant performance improvements across different scales, this computational overhead is reasonable and justified. Notably, the preprocessing phase (warmup + influence computation) is a one-time cost that can be amortized if multiple training runs are conducted.

---

### Decision · Action_Editor_5MYo · 2026-03-16

**Recommendation:** Accept with minor revision

**Additional Comments:**

There are additional results and arguments in the rebuttal that should go into the final version. In particular, I would like to see the clarification around DPO/NPO to go into the main text. This will strengthen the novelty of the paper. There are also a few wording issues (raised by reviewer sGcN) and typos (raised by reviewer NBXC). It would be good to have them fixed.

**Audience:**

Yes

**Audience Explanation:**

Fine-tuning large models will continue to be a main problem in this field.

**Claims And Evidence:**

Yes

**Claims Explanation:**

All reviewers agreed that the claims are supported by evidence. This is a more engineering paper, aiming at improving supervised fine-tuning. Perhaps the only claim of the paper is "Explicitly forgetting these tokens not only prevents the model from generating them but also helps avoid overfitting to the current corpus," which is not very different from saying the results improved.

There were some confusion around whether the proposed approach is simply a rebranding of NPO (raised by both reviewer NBXC and sGcN), but this was resolved during the rebuttal.

---

> ### Author Response · Authors · 2026-04-01
> **Acknowledgment and Thanks**
>
> Dear Editor and Reviewers,
>
> Thank you for your thoughtful feedback and the time you dedicated to evaluating our work. We have addressed the reviewer comments and applied the requested revisions to the manuscript. Your suggestions greatly helped us improve the quality of the paper.
>
> We look forward to contributing to the community through this work.
>
> Sincerely,
>
> Authors